# Pitfalls, Subtleties, and Techniques in Automata-Based Subword-Level Constrained Generation

**Marco Cognetta** [* 1]  **David Pohl** [* 1]  **Junyoung Lee** [2]  **Naoaki Okazaki** [1]

## Abstract

Constrained generation, where language models are forced to output text that adheres to a specified format, is a powerful tool for many tasks. Several libraries implement variants of it as the foundation for a larger feature set. In implementing our own version, we uncovered many subtle problems (some of which are present in existing libraries) that can affect the downstream performance of models that use constrained decoding.

Here, we describe common pitfalls and techniques when implementing robust constrained generation which apply to all major tokenizer families. Furthermore, we address favorable properties of our character-to-canonical pipeline (ease of use, efficiency, modularity, etc.). We hope this work guides you and your tokens to reliable, correct constrained outputs.[1]

## 1. Introduction and Background

Constrained generation has become a popular tool to enforce decoding constraints on LLMs and has found use across a variety of tasks such as question-answering, code generation, and information extraction. Several libraries implement constrained generation, with varying feature sets, trade-offs, and extractions. However, these are all optimized for production rather than experimentation and research.

In an ongoing work, we needed lower-level control over various aspects of constrained generation and implemented a lightweight framework. In doing so, we uncovered many subtleties that complicate constrained generation. These can have large downstream effects, so it is important to get them correct. Here, we cover some of our findings, both in terms

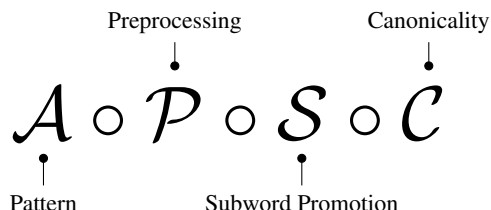

Figure 1: Our basic pipeline for converting user-defined, character-level constraints into subword-level constraints. $\mathcal{A}$ is the user-defined pattern. $\mathcal{P}$ is a set of preprocessing steps designed to mimic the tokenizer's pretokenization step or to add additional, modular constraints. $\mathcal{S}$ promotes character-level patterns to subword-level patterns. And $\mathcal{C}$ (optionally) filters out non-canonical subword sequences.

of edge cases and pitfalls, and also in terms of usability, extensibility, and performance. While we restrict our focus to regular constraints and the LLAMA2 tokenizer (which fit our research needs), the topics discussed here generalize to non-regular constraints and other tokenizer families.

We draw inspiration from speech processing literature (Mohri et al., 2008), which makes heavy use of modular, composed finite-state transducer and automata pipelines in order to produce a final model. This is particularly suitable given the recent connections that have been found between tokenization and automata theory (Song et al., 2021; Willard & Louf, 2023; Berglund et al., 2024; Cognetta & Okazaki, 2024; Koo et al., 2024). We focus on a finite-state pipeline that captures user-defined constraints (i.e., the desired output schema), tokenizer properties (e.g., the tokenization algorithm or byte-level fallbacks), and modular constraints (e.g., UTF-8 validity or mono- vs multi-linguality).

A summary of the main points of this paper is given in Table 1. We do not measure the effect of each of them individually here (which is highly sensitive to the task, prompt, and model), but note that they are common enough that we have observed each of them in existing production-level guided generation libraries,[2] and we have found that they impacted task performance in our concurrent research.

[1]Institute of Science Tokyo, Japan [2]Independent Researcher. Correspondence to: David Pohl <david.pohl@nlp.c.titech.ac.jp>, Marco Cognetta <cognetta.marco@gmail.com>.

[1]Our implementation can be found at https://github.com/mcognetta/constrainedgenerationpitfalls.

---

[2]A small list of related GITHUB issues is given in Appendix B.

| Section | Technique | Benefit | Tokenizer Dependent? |
|---|---|---|---|
| 3.1 | Tokenizer Preprocessing | Alignment with tokenizer configuration | Yes |
| 3.2 | Subword Promotion | Faster decoding (via constant time masking) 
 Alignment with tokenizer configuration | Yes |
| 3.3 | Canonical Filtering | Faster inference (by restricting search space) 
 Alignment with language model inductive prior | Yes |
| 3.3.1 | Coalescence | Faster inference (by skipping deterministic tokens) | Yes |
| 4.1 | Canonical Unicode | Prevents unrenderable outputs | No |
| 4.2 | Mono- vs Multi-linguality | Better modeling (via an additional semantic constraint) 
 Faster inference (by restricting search space) | No |
| 4.3 | Character Literals vs Bytes | Better modeling (by avoiding ambiguity) | Yes |
| 4.4 | Token Healing | Better modeling (by avoiding fragmented tokenizations) | No |
| 4.5 | Isolated Space Fusion | Better modeling (especially in Latin-script languages) | Yes |
| 5.1 | Marginalization | Better modeling (via well-grounded sampling) | No |

Table 1: A summary of the techniques in this paper, their benefits, and whether or not they are tokenizer agnostic.

We are not the first to implement constrained generation, even via automata theory. However, compared to Koo et al. (2024), who also describe a finite-state pipeline for constrained generation, we focus more on practical implementation pitfalls and optimizations. Furthermore, we attach an additional step to the pipeline, canonicalization, which can be precomputed and has favorable properties.

### 1.1. Automata and Transducers

We briefly overview the relevant aspects of automata theory but point the reader to (Riley et al., 2009) and (Sipser, 2013) for a more thorough treatment. Automata are a 5-tuple $\mathcal{A} = (Q, \Sigma, \delta, q_{\text{start}}, F)$ where $Q$ is a (finite) set of states, $\Sigma$ is a (finite) alphabet, $\delta$ is a transition function[3] $\delta : q \times \Sigma \to Q$, $q_{\text{start}} \in Q$ is the initial state, and $F \subseteq Q$ is a set of final states. Automata accept or reject strings over $\Sigma^*$. Let $w \in \Sigma^*$ be a string, then $w$ is accepted by $\mathcal{A}$ if there is a path $q_0, q_1, \ldots, q_n$ such that $q_0 = q_{\text{start}}$ and $\delta(q_i, w_i) = q_{i+1}$ and $q_n \in \mathcal{F}$. Let $\mathcal{L}(\mathcal{A})$ be the set of strings that are accepted by an automaton $\mathcal{A}$. This set is closed under many operations such as intersection ($\mathcal{L}(\mathcal{A}) \cap \mathcal{L}(\mathcal{A}')$), concatenation ($\mathcal{L}(\mathcal{A})\mathcal{L}(\mathcal{A}') = \{xy \mid x \in \mathcal{L}(\mathcal{A}) \lor y \in \mathcal{L}(\mathcal{A}')\}$), and Kleene-star ($\mathcal{L}(\mathcal{A})^* = \bigcup_{i=0}^{\infty} \mathcal{L}(\mathcal{A})^i$).

Finite-state transducers are a generalization of automata with an input and output alphabet. A transducer is a 6-tuple $\mathcal{T} = (Q, \Sigma, \Gamma, \delta, q_{\text{start}}, F)$ where $Q$, $\Sigma$, $q_{\text{start}}$, and $F$ are defined as with automata, $\Gamma$ is a (finite) output alphabet, and $\delta : Q \times \Sigma \cup \{\epsilon\} \times \Gamma \cup \{\epsilon\} \to Q$ is a transition function.[4]

---

[3]For simplicity, we restrict ourselves to *deterministic* automata, but all of our results extend to non-deterministic automata.

[4]We allow the empty string $\varepsilon$ on the inputs and outputs here as the input string and output string may be of different lengths, and since it is required for the subword promotion transducer (Cognetta & Okazaki, 2024; Koo et al., 2024). For simplicity, we omit them for automata, as they can always be made $\varepsilon$-free (Sipser, 2013).

Transducers accept *pairs* of strings $x, y \in \Sigma^* \times \Gamma^*$ if there is a path through the transducer such that the concatenation of the input and output symbols are $x$ and $y$, respectively.

Transducers share many of the closure properties of automata, but have two additional operations of interest. Let $\mathcal{T}_1$ and $\mathcal{T}_2$ be transducers with alphabets $(\Sigma, \Xi)$ and $(\Xi, \Gamma)$, respectively. Transducer composition is defined as $\mathcal{T}_1 \circ \mathcal{T}_2 = \{(x, y) \mid \exists z \in \Xi^* \text{s.t.} (x, z) \in \mathcal{L}(\mathcal{T}_1) \land (z, y) \in \mathcal{L}(\mathcal{T}_2)\}$. In other words, it transduces $x$ to $y$ through an intermediate string $z$. Automata and transducers can be composed (if $\Sigma = \Xi$), and automaton-automaton composition is the same as intersection. Transducer projection turns a transducer into an automaton accepting $\text{PROJ}(\mathcal{T}) = \{y \mid (x, y) \in \mathcal{L}(\mathcal{T})\}$.

### 1.2. Constrained Generation

Constrained generation is a collection of techniques to force language models to output sequences that match a specified pattern by essentially setting the probability of invalid sequences to zero. The standard way to do this is to mask out logits corresponding to invalid tokens before sampling during generation. That is, instead of sampling a token $t_i$ given a context $c_l$ from a softmax distribution

$$p(t_i \mid c_l) = \text{SOFTMAX}(l)_i, \quad (1)$$

where $l \in \mathbb{R}^{|\Gamma|}$ are the logits that are produced by a language model for $c_l$, we sample from

$$p'(t_i \mid c_l) = \text{SOFTMAX}(l \odot m)_i, \quad (2)$$

where $\odot$ is the Hadamard product and $m \in \mathbb{R}^{|\Gamma|}$ is a mask

$$m_i = \begin{cases} 1 & v_i \in \Gamma \text{ can satisfy the constraint} \\ -\infty & \text{otherwise.} \end{cases}$$

The masking ensures that tokens that cannot satisfy the constraint are given probability 0 after the SOFTMAX operation.

The unconstrained probability of a sequence $t = t_1 t_2 \ldots t_n \in \Gamma^*$ can be computed autoregressively as

$$p(t) = p(t_1 \mid \texttt{}) \prod_{i=1}^{|t|-1} p(t_{i+1} \mid \texttt{} t_1 t_2 \ldots t_i). \quad (3)$$

Likewise, the constrained probability is typically given by

$$p'(t) = p'(t_1 \mid \texttt{}) \prod_{i=1}^{|t|-1} p'(t_{i+1} \mid \texttt{} t_1 t_2 \ldots t_i), \quad (4)$$

though we discuss alternative formulations in Section 5.1.

Following the autoregressive nature of modern language models, we can check if the sequence resulting from adding a given token to the current context is a prefix of a sequence that matches the constraint. Let $t_1 t_2 \ldots t_k$ be some context and $v_i$ be the token corresponding to the logit $l_i$, then $m_i = 1$ if $t_1 t_2 \ldots t_k v_i \Gamma^* \subseteq \mathcal{L}(\mathcal{A}_s)$, where $\mathcal{A}_s$ is a subword-level automaton (e.g., the result of the equation in Figure 1).

## 2. Constrained Generation Packages

### 2.1. Outlines

OUTLINES (Willard & Louf, 2023) is a fully-featured constrained generation library. They provide regular and context-free constraints built into a prompt-templating system. To our knowledge, OUTLINES was the first to perform subword promotion, which they implement as an "index". Rather than a purely finite-state approach, OUTLINES allows users to define a constraint (we refer to regular constraints here, but the same is true for context-free constraints) which is converted to an automaton. They then iterate over each state of the automaton and determine, for each token in the vocabulary, what state (if any) it would lead to if read character-by-character from the current state. This produces a hash table that allows one to traverse the character-level automaton using subword tokens. Additionally, it produces a static set of valid subword tokens for each state.

On top of this foundation, they build a large feature set for common constrained generation use cases like JSON output or categorical sampling and includes other generic LLM inference techniques like chain-of-thought.

### 2.2. Guidance

GUIDANCE (Guidance AI, 2023) is another fully-featured constrained generation library. Like OUTLINES, it allows for regular and context-free constraints but has additional functionality for *dynamic* constraints—constraints can be added, removed, or modified during inference, depending on prior output.

Perhaps partially due to dynamic constraints, GUIDANCE does not pre-compile a subword-level constraint. Instead, it

dynamically constructs masks by iterating over each token in the vocabulary and checking if it can satisfy the final constraint. The authors argue that there is a beneficial trade-off between the time it takes to perform dynamic masking and the preprocessing time required to construct a compiled, static mask (Geng et al., 2025).

GUIDANCE makes use of an abstract syntax tree data structure for representing constraints and, in this way, allows for modularity (i.e., by defining a `GrammarNode` which corresponds to a family of constraints.).

### 2.3. XGrammar

XGRAMMAR (Dong et al., 2025) is the most similar in scope to our implementation. Compared to OUTLINES and GUIDANCE, it has relatively few features and rather focuses on being a highly-performant constrained generation core that other functionality can be built on.

Like GUIDANCE, it uses dynamic masking built on top of a custom grammar framework but mixes static and dynamic constraints with an "adaptive token mask cache". This cache identifies when tokens at a specific rule in a grammar are "context-independent" and can be immediately verified without moving into nested rules. They find such tokens are relatively common and can be cached so that they don't need to be recomputed later. They additionally provide many other decoding time optimizations to ensure that dynamic masking is performant, such as token prefilling.

### 2.4. Other

Two popular generic inference libraries, LLAMA.CPP (Gerganov & Community, 2023) and VLLM (Kwon et al., 2023), include basic structured generation as a feature. LLAMA.CPP implements their own, which uses dynamic masking, while VLLM integrates OUTLINES and GUIDANCE. SGLANG is another inference engine designed for constrained generation (Zheng et al., 2024). They provide a number of constraint primitives, inference primitives, and runtime optimizations to enable efficient constrained generation. DOMINO (Beurer-Kellner et al., 2024) is a framework that focuses on performant and "minimally-invasive" constrained generation with dynamic masking.

### 2.5. Dynamic vs. Precompiled Constraints

Constrained generation libraries are split over how to treat masking, with some (like OUTLINES and (Koo et al., 2024)) precompiling static masks that can just be looked up at will, and others GUIDANCE and DOMINO performing on-the-fly, dynamic masking, where at each inference step, the entire vocabulary is enumerated and checked for validity. XGRAMMAR mixes the two by selectively caching masks where possible and dynamically computing them otherwise.

A clear consensus on which is better has not been reached, with proponents of dynamic masking arguing that it is faster (no start-up time and relatively low runtime overhead, especially when compared to the runtime cost of the base language model's inference step) and more flexible (Geng et al., 2025; Beurer-Kellner et al., 2024; Dong et al., 2025), while precomputed-masking proponents argue it is faster when amortized over the life of the model (pay a start-up cost for optimized inference) (Koo et al., 2024), especially as the size of modern LLM vocabularies grow (e.g., the GEMMA3 tokenizer has 260k elements in the subword vocabulary (Gemma Team et al., 2025)).

Like Willard & Louf (2023) and Koo et al. (2024), we employ static constraints based on compiled automata. Static constraints fit nicely into our framework as the combination of many modular automata is itself an automaton that can be optimized for efficient usage during inference. However, many of the techniques we present here also apply to dynamic-masking constrained generation.

## 3. Compiling Automata From Tokenizers

We now focus on how to compile automata from tokenizer configurations to enable constrained generation that matches the implementation details of the tokenizer.

### 3.1. Preprocessing

One difficult aspect of designing proper constraints is that tokenizers often have a preprocessing normalization step that transforms the raw text input into a standardized format. A problem arises where the user must a) know the normalization requirements of a particular tokenizer and b) be able to write their constraints in a way that matches it.

On the other hand, using a finite-state pipeline, this can be hidden from the user. After the user has defined a pattern (without knowledge of the pretokenization steps), we can use a precompiled preprocessing transducer to implement the pretokenization and convert the user's pattern into one that matches what the text would actually look like during inference. By abstracting this away from the end-user, we reduce the likelihood of errors in the constraint definition and, by making tokenizer-aware pipelines, we allow for per-tokenizer, precompiled preprocessing transducers to be constructed, which centralizes the sources of errors.

#### 3.1.1. EXAMPLE: START-OF-SEQUENCE AND SPACING

The example in Figure 2(a) contains a `Prepend` preprocessing step, where a space is added to the very start of the input sequence,[5] and space tokens are converted to a

---

[5]This is because the LLAMA2 tokenizer is built on SENTENCE-PIECE, which has this behavior by default.

```
"normalizer": {
  "type": "Sequence",
  "normalizers": [
    {
      "type": "Prepend",
      "prepend": "__"
    },
    {
      "type": "Replace",
      "pattern": {
        "String": " "
      },
      "content": "__"
    }
  ]
},
"pre_tokenizer": null,
```

(a) The preprocessing step for LLAMA2.

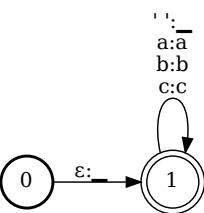

(b) The preprocessing transducer.

Figure 2: The preprocessor for LLAMA2 and a transducer which implements it. Composing this with the input pattern preprocesses it to match the tokenizer specification.

special space marker "__" (U+2581). This can be realized by a finite-state transducer, shown in Figure 2(b).

The example shown here is simple (by choice, for space reasons), but arbitrarily complex preprocessors could exist that handle things like text normalization (e.g., punctuation normalization) or conversion (e.g., canonical forms for numbers or addresses) (Ebden & Sproat, 2015).

### 3.2. Subword Promotion

Subword promotion promotes character-level patterns to subword-level patterns—in other words, it accepts subword sequences $t \in \Gamma^*$ if the detokenized form matches the character-level constraint $\mathcal{A}$. This allows the users to encode their constraints in a standard (character-level) regex, which abstracts away things like the subword vocabulary of the model they are using and subword boundary marking schemes. This step additionally allows for $O(1)$-time satisfiability queries—that is, given a constraint, a context, and a subword token, we can determine in constant time whether the token, when added to the context, can still possibly satisfy the constraint. This is in contrast to "on-the-fly" checks like those used in XGRAMMAR and GUIDANCE.

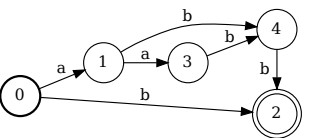

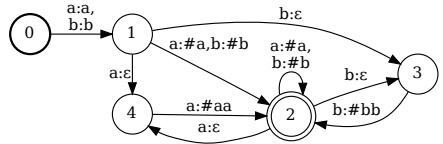

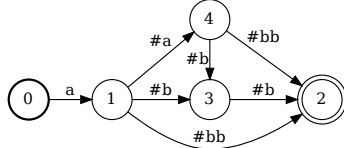

(a) A character-level automaton.   (b) A character-to-subword transducer.   (c) The promoted subword-level automaton.

Figure 3: An example of subword promotion. Composing the automaton (a) with the transducer (b) produces a new automaton (c) where each path defines a subword sequence that would match the original (a) at the character level.

In the on-the-fly setting, the satisfiability query is done with the original, character-level pattern — when checking if a token can satisfy the constraints, the token is read out character-by-character and the constraint automaton is traversed to ensure that it is potentially satisfying. This avoids a potential startup cost (constructing the compiled, subword-level pattern), but introduces runtime overhead (scanning the vocabulary and processing each token) (Geng et al., 2025).

In our finite-state pipeline, subword promotion is done by building a character-to-lexicon transducer that transduces character sequences to subword tokens (Cognetta & Okazaki, 2024; Koo et al., 2024). Specifically, it encodes the language $\cup_{w\in\Gamma}([\text{w}_1][\text{w}_2]\ldots[\text{w}_n],[\text{w}])$ and is composed with the input pattern then projected to an automaton over subwords. Figure 3 gives an example of subword promotion.

### 3.3. Canonical Filtering

The subword-promotion transducer promotes character-level patterns to subword-level patterns that accept *any* sequence satisfying the character-level pattern. For example, if the constraint allowed the word "Alphabet", the subword automaton would accept [Al][pha][bet], [Alphabet], [Alpha][bet], etc.

This behavior does not reflect that a tokenizer is a one-to-one mapping from strings to token sequences, so a natural extension is to allow only the sequences the tokenizer would produce. Let $T : \Sigma^* \to \Gamma^*$ be a tokenizer, and $T^{-1}$ be the inverse, detokenization function. Tokenizers define one-to-one mappings, but, the character-to-subword transducer $\mathcal{S}$ is a one-to-many mapping that allows any sequence $t'$ such that $T^{-1}(t) \in \mathcal{L}(\mathcal{A})$. Instead, we want to restrict the subword automaton to only sequences $t$ such that $T^{-1}(t) \in \mathcal{L}(\mathcal{A})$ and $T(T^{-1}(t)) = t$, the *canonical tokenization*.

Previous works (Song et al., 2021; Berglund et al., 2024; Cognetta & Okazaki, 2024; Tran-Thien, 2024) show that the automaton of only canonical tokenizations can be pre-computed, which is costly at first, but only has to be done once for a tokenizer. Consequently, ensuring canonicality

is as simple as intersecting the subword-level automaton with the canonical automaton. Figure 4 shows a subword-level automaton and the remaining paths after filtering out non-canonical sequences.

Limiting the paths in our subwords-level automaton to only those corresponding to canonical tokenizations is beneficial for several reasons. As per the *stars and bars theorem* (Wikipedia contributors, 2025), the number of ways to tokenize a sequence of length $n$ into $k$ tokens is $\binom{n-1}{k-1}$, so the total number of ways to tokenize a string $s$ is (at most)

$$\sum_{k=1}^{|s|} \binom{|s|-1}{k-1} = 2^{|s|-1}. \tag{5}$$

On the other hand, there is only one canonical tokenization of $s$, which massively reduces the number of acceptable strings in our subword automaton when combined with a canonical constraint filter.

Furthermore, ensuring canonicality maintains the inductive bias of our model, as canonical tokenizations are what the model has been trained on. Lastly, as described in the next section, decoding speed is vastly improved, especially when combined with a constraint pattern.

Canonical filtering has a downside in that the size of the resulting automaton can be very large compared to the character-level automaton (Berglund et al., 2024; Cognetta & Okazaki, 2024), especially if repetition operations are involved. We have found that, unless the constraint is heavily restricted, using canonical filtering with repetition results in automata that are too large to be used efficiently. We have found restricting the search space (e.g., by specifying a set word list, rather than all strings over $\Sigma$) and using $\Sigma^*$ rather than $\Sigma^{\leq k}$ (where applicable) helps manage this to a degree.

### 3.3.1. SPEEDING UP DECODING ("COALESCENCE")

A speedup in decoding is based on the observation that, for a given state in the constraint automaton, decoding can be skipped if there is only one outgoing arc (since there

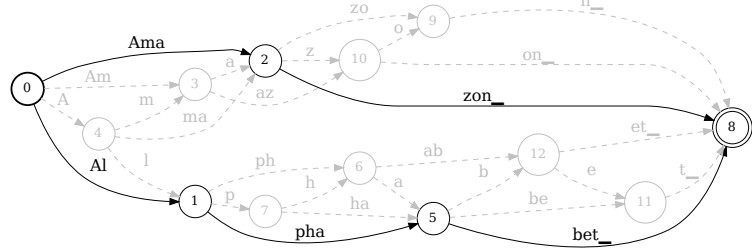

Figure 4: A subword-level constraint for the words {Alphabet, Amazon}. The darker paths represent the canonical tokenizations (Section 3.3). Other than at the starting state, each state has at most one canonical arc. Thus, once we have decoded the first token, we do not need to actually decode the rest and can simply add them to the context (Section 3.3.1).

is only one valid token, which must have probability 1). As canonicality filtering removes all alternative paths that do not match the canonical tokenization, this situation is encountered frequently, especially for a restrictive pattern like a multiple-choice constraint (see Figure 4).

With JSON generation being a major application of constrained generation, we conduct a small experiment to show the speed-up generated by canonical filtering. We construct a prompt and a simple JSON template to generate Pokedex entries for Pokemon. We create 100 JSON entries and measure the average time per entry and the ratio of skippable tokens across all generation steps.

| Model | Canonical | Skip Rate | Rel. Time |
|---|---|---|---|
| LLAMA2-7B | no | 24.5% | - |
| | yes | 77.9% | -15.7% |
| LLAMA2-13B | no | 26.5% | - |
| | yes | 78.9% | -21.0% |

Table 2: Effect of decoding-skipping in combination with canonicality on inference time. Note how canonicality notably increases the skip rate and decreases inference time. The JSON schema is given in Appendix C.

OUTLINES proposed the idea of "coalescence", in which an ad-hoc heuristic is described to find and remove such detours in the decoding automaton,[6] and a similar technique was described by Tran-Thien (2024) and Zheng et al. (2024). However, the OUTLINES proposal does not guarantee finding all such optimizations nor does it guarantee the remaining paths correspond to the canonical tokenization. Our implementation of canonicality as a finite-state transduction (Cognetta & Okazaki, 2024) guarantees all optimizations naturally without additional effort.

# 4. Pattern Design and Modular Filtering

Now that our core pipeline is implemented, we turn to various modular constraints that can be immediately integrated into the pipeline via a simple composition so that the user does not have to implement it themselves or even know the details in depth. These constraints are largely orthogonal to the correctness of the user-defined scheme or the integration with the tokenization specification, but can be used to improve modeling or runtime performance or to simply constrain the output in a more fine-grained way without resorting to writing extremely complex regular expressions.

## 4.1. Canonical Unicode

Often, LLMs use a (UTF-8) byte-level fallback to ensure that all sequences are representable by the tokenizer. When generating text, the model may fall back to generating bytes (which will be converted to Unicode codepoints during detokenization), which introduces the possibility of generating invalid byte sequences. We can place a "canonical UTF-8" automaton in our finite-state pipeline that guarantees only valid UTF-8 sequences are generated (Höhrmann, 2010).

Unicode canonicality applies not only to the UTF-8 byte-level sequences, but also to the sequence of Unicode codepoints (i.e., the equivalence between two distinct codepoint sequences that represent the same character). For example, when a sequence of codepoints is used to represent a character that has its own codepoint representation — for example, u+° (U+0075 and U+030A) vs ů (U+016F). This many-to-one phenomenon can also be observed in South Asian Brahmic scripts, where finite-state processing has been successfully employed to perform canonical normalization and well-formedness checking of orthographic syllables in Unicode (Johny et al., 2021; Gutkin et al., 2022).

Enforcing this version of Unicode canonicality via automata can be easily integrated into our pipeline as a modular con-

---

[6]https://blog.dottxt.co/coalescence.html

straint, without requiring the user to encode it manually into their initial constraint (which would be very difficult).

### 4.2. Mono- vs Multi-linguality Filtering

Modern LLMs are typically trained on multilingual corpora (which is reflected in their tokenizer vocabulary). However, it is often the case that inference is done in a monolingual setting (e.g., question-answering tasks tend to be done in a single language, at least at the individual question level). Allowing multilinguality when monolinguality is desired has two negative effects: 1) probability mass is guaranteed to leak from the desired language to tokens of other languages and 2) a larger space of acceptable sequences is available, which slows decoding.

Aligned with one of our central goals of making it easier for users to define a prompt and have it be refined later, we want to alleviate the burden of properly restricting their initial prompt to the correct language. We can introduce *monolinguality* filters for this, which are placed between the constraint automaton and the subword-promotion transducer to filter out character sequences that do not match the intended language (e.g., by using the language script as a proxy). For example, we could use ASCII roughly as a proxy for English or the Korean Hangul Precomposed Syllable range for Korean, etc. One could even just use a dictionary encoded as a trie to restrict to only known words in a given language.

These automata are deterministic, individually small, tokenizer agnostic, can be precomputed, etc., meaning that they can all be prepared ahead of time by the library maintainer and used in a modular fashion (for example, by the user specifying an `acceptable_scripts` parameter).

### 4.3. Ambiguity Between Characters and Bytes

Modern tokenizers often have implicit byte-level fallbacks in order to handle the wide variety of scripts that arise in a multilingual setting. If a character (i.e., an *atomic* token, one which is *not* formed by a merge in BPE) is not present in the vocab but is seen in training, its Unicode byte representation is used and the character is modeled by a special byte sequence. For example, in Korean, the common character 한 might be present in a multilingal tokenizer's vocabulary, but the rare 궭 might not. During encoding these would be encoded as [한] and [<0xEA>][<0xB6>][<0xAD>], respectively (with the latter being three subword tokens).

The full set of 256 bytes is included in the subword tokenizer's vocabulary, which adds an ambiguity between characters (e.g., a, b, c) and ASCII bytes (e.g., <0x61>, <0x62>, <0x63>). Not accounting for this ambiguity means that during generation, the model could produce the ASCII-byte representation of a character instead of the lit-

eral character, thereby polluting the modeling context and harming the downstream performance of the model. We can filter this out by adding an additional filter automaton to the pipeline that disallows byte sequences (where applicable) or disallows ASCII bytes (i.e., it allows non-ASCII UTF-8 sequences, but forces ASCII to use the character literals).

This ambiguity is present in OUTLINES, which allows generation of ASCII sequences through their byte values (or a mix of byte and character literals).

### 4.4. Token Healing

One potential issue comes at the pattern-design level, especially where the prompt ends and the constraint begins. This has been observed before in GUIDANCE, where it is referred to as *token healing*.[7] They recommend a simple procedure where, given a prompt with a constraint at the end, they first backtrack a single token and add the surface form of the final token of the prompt to the pattern.

The backtracking technique is orthogonal to the rest of the constrained generation system, so it can be integrated into the system we describe in this paper. However, there is an alternative way to implement it, with essentially no runtime overhead, and while guaranteeing that token boundary issues are eliminated, by using canonical filtering (Section 3.3).

Until this point, we have only considered constraint specification, subword promotion, and filtering as something separate from the prompt. We could instead integrate the prompt into the constraint directly and treat it as a prefix to the regular language that actually encodes the constraints (that is, we combine the prompt and the constraint via concatenation). Then, we continue with the rest of the finite-state pipeline, including canonical filtering.

As canonical tokenization is a one-to-one mapping with the input text, the first part of the canonical promoted subword automaton (corresponding to the static prompt) is entirely deterministic. Using the efficient decoding method in Section 3.3.1, we can simply skip decoding during this entire section of the automaton before reaching the constraint boundary. Since we have applied canonical subword filtering, the boundary between the prompt and the constraint is guaranteed to be well-formed, and we can begin decoding without any explicit token-healing backtracking.

### 4.5. Isolated Spaces

One issue that arises from subword promotion is isolated spaces. Due to the requirement of an open vocabulary, there is an individual space character present in the vocabulary (in

---

[7]https://guidance.readthedocs.io/en/latest/example_notebooks/tutorials/token_healing.html

fact, there tends to be two: the byte-level ASCII character `<0x20>` and the literal space character).

A problem can arise where subword sequences which do not fuse the leading space and eventual word prefix (for example, `[␣␣sevent][een]` vs `[␣␣][s][event][een]`) are sampled, leading to poor downstream performance. Table 3 shows examples where the non-fused space tokenization has much lower probability, but was generated because sampling an isolated `[␣␣]` had high probability.[8] Eliminating non-fused tokenizations when appropriate[9] eliminates a subtle source of error due to poor tokenizations.

| Prompt | Token 1 | Token 2 | Token 3 | Token 4 | Sequence | Matches Constraint? |
|---|---|---|---|---|---|---|
| | $-2.57$ [␣␣sevent] | $-9e{-}4$ [een] | | | $-2.58$ "seventeen" | ✓ |
| In Roman numerals, XVII means | $-0.09$ [␣␣] | $-5.77$ [se] | $-2e{-}4$ [vent] | $-2e{-}3$ [een] | $-5.87$ "seventeen" | ✓ |
| | $-0.09$ [␣␣] | $-9.75$ [s] | $-5.41$ [event] | $-0.03$ [een] | $-15.29$ "seventeen" | ✓ |
| | $-0.09$ [␣␣] | $-\infty$ [1] | $-\infty$ [7] | | $-\infty$ "17" | ✗ |

Table 3: Tokenizations and log probabilities of some sequences (using LLAMA2-7B). The high probability of the isolated space `[␣␣]` (due to `[␣␣][1][7]` having high probability in the unconstrained setting) means it is likely to be sampled. However, if the constraint is `[A-Za-z]*`, we fall into a subspace that generates low probability, non-canonical tokenizations of "seventeen".

# 5. Constraint Inference

Let $\mathcal{A}$ be some constraint. Equation 4 attempts to approximate the joint probability distribution $p(t, \mathcal{A})$, where a sequence $t$ is given probability 0 if it does not match the constraint. However, this distribution is malformed, and, in reality, the partition function $Z = \sum_{t \in \mathcal{L}(\mathcal{A})} p(t)$ must be computed in order to model the true distribution $p(t, \mathcal{A}) = \frac{p(t)}{Z}$.

## 5.1. Constraint Marginalization

Let $\mathcal{A}$ be a character constraint and $\mathcal{A}_s$ be the subword-promoted version of it. Computing the value of $Z$ for $\mathcal{A}_s$ is difficult as $\mathcal{L}(\mathcal{A}_s)$ could be infinite (and even in the finite case, Equation 5 shows that the size of the language could be exponentially larger than $\mathcal{L}(\mathcal{A})$).

Instead, we can use importance sampling to compute the

---

[8]The reason that the tokenization `[␣␣][se][vent][een]` has such high (relative) probability is that that is the tokenization produced for "seventeen" (including the quotes) where no leading space marker is added. Thus, this is the canonical, non-leading-space tokenization and likely appears in the training data.

[9]We note that, depending on the tokenizer, token fusion is not always done. For example, LLAMA2 tokenizes "Korea" to `[␣␣Korea]` but 한국 to `[␣␣][한][국]`.

---

marginal probability by sampling from a simpler distribution[10] (Geh et al., 2024). Let $q$ be a proxy distribution for which we can easily sample sequences that match $\mathcal{A}_s$. Then,

$$p(\mathcal{L}(\mathcal{A}_s)) = \sum_{t \in \mathcal{L}(\mathcal{A}_s)} p(t) = \mathbb{E}_{t \sim q}\left[\frac{p(t)}{q(t)}\right] \approx \frac{1}{N}\sum_{i=1}^{N} \frac{p(t^{(i)})}{q(t^{(i)})},$$

where $t^{(i)}$ is sampled from $q$. By using $p'$ (Equation 4) as the proxy distribution, we guarantee that we can sample from $\mathcal{L}(\mathcal{A}_s)$, since that is exactly the support of $p'$.

To our knowledge, all of the current constrained generation libraries implement a variety of samplers (e.g., greedy, beam search, speculative decoding, etc.), but none of them account for the malformedness of the distribution induced by logit masking as described in Equation 2 or provide constraint marginalization as a feature. Our implementation exposes the raw (un)constrained logits, and we implement constraint marginalization on top of it. In some cases, especially when using canonical filtering (Section 3.3), exact marginalization is tractable, and our library supports that as well.

## 5.2. Other Inference Methods

Marginalization is slow using an enumerative method, and importance sampling can take a long time to converge (Geh et al., 2024). Other methods for sampling from complex distributions exist, such as sequential Monte Carlo or adaptive rejection sampling (Lipkin et al., 2025). Recent work has considered training and inference strategies for ensuring canonicality (Vieira et al., 2025). In addition to this, the exact formulation of the marginalization plays a role in correctness. In particular, Pimentel and Meister (2024) discuss the nuance of marginalizing subword sequences to compute the probability of a word when the tokenizer uses beginning-of-word markings (e.g., `[␣␣token][s]`) compared to end-of-word markings (e.g., `[token][s␣␣]`).

# 6. Overview of Our Package

Our package is designed to be a lightweight foundation for constrained generation research. We do not aim to compete with popular packages, which are created for use in production and offer many utilities and functions for downstream tasks. We design our package for 1) simplicity, 2) control, and 3) convenience. Briefly, we provide a simple, minimal interface compatible with HuggingFace models; generation of (constrained and unconstrained) probabilities; tokenizer preprocessing, pattern filtering, and canonicalization; coalescence; and constraint marginalization.

---

[10]We can sample from $p$ directly, but if the probability of sequences that match $s$ is small, we will mostly sample invalid strings, so the convergence will be slow.

```
# Build pipeline from a HF causal model
pipe = Pipeline("meta-llama/Llama-2-7b-hf")

# Restrict output to ASCII characters only
pipe.restrict_characters_to("ascii")

# Multiple-choice constraint (with canonicality)
cities = Constraint(
    regex="Cologne|Berlin|Munich", canonical=True
)

prompt = "What is the capital of Germany?"

sampled = pipe.generate(prompt, constraint=cities)
```

```
# "Berlin"-only constraint (any tokenization)
berlin = Constraint("Berlin")

# Compute probability mass of constraints
total_mass = pipe.compute_probability_mass(
    prompt, cities
)

berlin_mass = pipe.compute_probability_mass(
    prompt, berlin
)

# Compute normalized, marginal log prob of Berlin
berlin_marginal = berlin_mass - total_mass
```

Figure 5: A basic example of our implementation's interface. The user provides a model, prompt, and constraint and then selects various modular filters and sampling strategies. Everything is composed nicely using our finite-state framework, allowing the user to perform efficient inference for constrained generation.

### 6.1. Ease-of-Implementation and Verification

Our goal with this project was to build a functional core for constrained generation research, not a fully-featured guided generation library. As such, our goals were rapid and confident iteration while retaining relatively high performance.[11] As noted by Koo et al. (2024), using automata theory as our base abstraction enables all of these properties, as automata are well-understood, capture the expressiveness of the constraints that we want to implement, and have many high-quality and performant libraries.

For our regular expression engine, we used INTEREGU-LAR[12] (which is also what OUTLINES uses). We used PYNINI (Gorman, 2016), a Python library built on top of OPENFST (Riley et al., 2009), a well-tested C++ library for finite-state transducers, for implementing the various modular constraints. By offloading the underlying algorithmic code to OPENFST, we effectively eliminate a large class of implementation errors, allowing us to focus on more relevant aspects of constrained generation. This is in contrast to the current major constrained generation libraries, which frequently deal with subtle parsing bugs.

Our inference engine is built on the basic TRANSFORMERS interface (Wolf et al., 2020), where we simply intercept the logits during generation to apply our masks.

In total, our implementation (including all modular constraint definitions, the process to build a canonical tokenization automaton, the inference engine, etc.) is roughly 1k lines of Python code. On the other hand, XGRAMMAR uses 12k lines of C++ (we note that they implement more features, but the core of their code is their grammar parsing and inference engine) (Dong et al., 2025, Section 4).

### 6.2. Example Usage

Here, we provide example code snippets for the core interface of our implementation. In Example 5, we detail a basic constrained generation inference run, where a user can define the relevant parameters (model, constraints, optional filters, and decoding strategies) and show how to use it for constraint marginalization, as described in Section 5.1.

The interface is deliberately simple, and is designed to give users easy access to the internals of the system and return (constrained or unconstrained, and optionally marginalized) sequence probabilities under specific constraints, which form the basis of all other language modeling applications. Thus, this simple interface can be used as the core for more feature-rich constrained generation libraries.

## 7. Conclusion

Constrained generation has proven to be a powerful tool for large language models, but they are not without downsides. The intersection between the model's decoding algorithms and user-defined constraints requires care to properly implement, and not doing so can introduce difficult-to-detect bugs. By using well-defined and verifiable constraints with well-tested libraries, those implementing constrained generation are able to side-step many of these issues without sacrificing performance.

Our work focused on common but subtle pitfalls that can affect constrained generation systems. For each of these, we showed how a well-grounded automata-theoretic framework can handle them elegantly and efficiently. Many of these pitfalls exist in popular constrained generation libraries and can be easily overlooked. We prepared a small implementation of our ideas, which we hope aids future researchers in implementing constrained generation.

---

[11]These are also reasonable goals for production environments.
[12]https://github.com/MegaIng/interegular

## Acknowledgments

These research results were obtained from the commissioned research (No.22501) by National Institute of Information and Communications Technology (NICT), Japan. This work was supported by JSPS KAKENHI Grant Number 25H01137. David Pohl is supported by fellowships of the German Academic Exchange Service (DAAD) and the German Academic Scholarship Foundation (Studienstiftung).

## Impact Statement

This paper presents work whose goal is to advance the field of Machine Learning. There are many potential societal consequences of our work, none which we feel must be specifically highlighted here.

## Limitations

Our work provides an example implementation based on the LLAMA2 BPE tokenizer, which is a relatively small model with a small vocabulary size compared to state-of-the-art models but still fits our research needs succinctly.

As modeling performance of constrained generation on any downstream task is highly model-dependent, we do not include empirical evaluations that draw away from the focus of the work on implementation.

Context-free grammars were not in the scope of our ongoing work and, hence, our research needs, and is not included for simplicity of presentation.

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

# A. More About Our Package

## A.1. Pipeline

All features are available through the `Pipeline` class, which encapsulates a `model` and a `tokenizer` from the TRANSFORMERS library.

```python
from constrain import Pipeline

pipeline = Pipeline("meta-llama/Llama-2-7b-hf")
```

## A.2. Constraint

Constraints are represented by the `Constraint` class, which is instantiated from either a regular expression or a finite-state machine. Enforcing canonicality is as simple as enabling the `canonical` option.

```python
from constrain import Constraint

# "What is the capital of Germany?"

# Enforce one of the given options and canonicality
constraint = Constraint(regex="Cologne|Berlin|Bonn|Munich", canonical=True)
```

More complex cases may be hard to express in the form of a regular expression and transforming an automaton to regex is costly. Therefore, the `Constraint` accepts an automaton directly. By default, our package supports `interegular` automata, but the engine can be swapped out by subclassing the `FSMAdapter`.

```python
from interegular import parse_pattern

# "What was the capital of Germany before 1990?"

berlin = parse_pattern(".*Berlin.*").to_fsm()
anything = parse_pattern(".*").to_fsm()

ban_berlin = anything - berlin

constraint = Constraint(fsm=ban_berlin)
```

## A.3. Functions

The `generate` function expects a prompt or a list of prompts and supports the common sampling parameters `do_sample`, `top_k`, `top_p`, `temperature`, `max_new_tokens`.

```python
pipeline.generate(
    "What is the capital of Germany?",
    do_sample=True,
    temperature=0.8
)
```

For instruction-tuned models, we should follow the model-specific pattern, which is provided as a convenient function in the TRANSFORMERS library with `apply_chat_template`. One issue with leaving the chat templating to the user is that different templates behave differently, and some automatically add special tokens. If we expect the prompts to be strings, checking whether special tokens are already part of the string is complicated. Therefore, all functions accept message dictionaries, to which the template is internally applied.

```python
messages = [
    {"role": "system", "content": "You are a helpful assistant."},
    {"role": "user", "content": "What is the capital of Germany?"},
    {"role": "assistant", "content": "Answer:"}
]

pipeline.generate(messages)
```

By passing a `Constraint` object, the prompt or prompts are constrained.

```
cities = Constraint(
    regex="Berlin|Frankfurt"
)

pipeline.generate(
    "What is the capital of Germany?",
    constraint=cities
)
```

To analyze alternative continuations, i.e., "how would the model proceed if the first sampled tokens were $t_1 t_2 \ldots t_k$", we can pass the `response_prefix` parameter.

```
tokens = ["_", "B"]
# OR
tokens = [29871, 29933]

pipeline.generate(
    "What is the capital of Germany?",
    response_prefix=tokens
)
```

As a core principle, all functions return log-probabilities. That means, for the above generation, we are not just provided with the decoded output, but also with step-wise and cumulative log probabilities both for the unconstrained and constrained scenario. Returning the probabilities is not only helpful for a researcher analyzing hand-selected scenarios, but also serves as a starting point for computing the marginal probability of a string. The `compute_probability_mass` function returns the total mass of a constraint for a given prompt.

```
{
  "strings": ["Berlin"],
  "tokens": {
    "ids": [[2292, 1915]],
    "strs": [["_Ber", "lin"]],
  },
  "log_probs": {
    "unconstrained": {
        "per_token": [[-1.2, -0.2]],
        "cumulative": [-1.4],
    },
    "constrained": {
        "per_token": [[-0.6, -0.1]],
        "cumulative": [-0.7],
    }
  }
}
```

By passing token sequences to `compute_probability`, we obtain the probabilities of each sequence, constrained and unconstrained. This is used internally for marginalizing over canonical constraints, as they are enumerable and do not require costly sampling.

```
pipeline.compute_probability(
    prompt, tokens=[["_Col", "ogne"]]
)

# output
{
    "unconstrained": {
        "per_token": ...,
        "cumulative": ...,
    },
    "constrained": {
        "per_token": ...,
        "cumulative": ...,
    },
}
```

## A.4. Character Restriction

As described in Sections 3.3 and 4.2, it can be useful to restrict the characters to a certain subset in order to concentrate probability mass in the desired output tokens and speed up automaton traversal and decoding. The allowed characters can be globally restricted to a custom set of characters, which propagates through tokenization and transduction.

```
pipeline.restrict_characters_to("hangul")
# OR
pipeline.restrict_characters_to(
    chars=["a...zäöüß"]
)
```

# B. Example Issues in Existing Works

| Package | Issue Title (Linked) | Summary | Relevant Section |
|---------|---------------------|---------|------------------|
| OUTLINES | Constrained generation does not account for tokenizers prepending special symbols | The constrained generation mechanism in OUTLINES fails to produce canonical tokenizations when using tokenizers that prepend special symbols (e.g., SENTENCEPIECE-based tokenizers like LLAMA and PHI). | 3.1; This is an issue with tokenizer preprocessing. |
| GUIDANCE | Regex in guidance.gen fails to handle non-ASCII characters like German umlauts | Regex rules in guidance.gen fail to handle non-ASCII characters (e.g., German umlauts such as ä, ö, ü, ß). Even when explicitly included in the regex pattern, the generated text systematically omits these characters. | 4.1; This is a rough edge between the tokenizer and parser implementations. |
| XGRAMMAR | MaxLength not respected | The maxLength constraint specified in the JSON schema is not enforced. | 3.2; This is made trivial in our framework by adding a modular constraint of $\Gamma^k$. |
| | Unable to match unicode chars | Unicode characters are not correctly handled, and attempts to define grammar rules that accept a wide Unicode range (e.g., [a-zA-Zà-ÿÀ-Ö0-9]) fail to match expected inputs, even with wildcard patterns. | 4.1. This is a rough edge between the tokenizer and parser implementations. |

Table 4: A small list of representative errors that are caused by the difficulties of properly implementing parsers, tokenizers, and complicated schema (for the users). Many of these are trivially fixed by modular finite-state constraints but are not easily integrated into custom parsers that do not use an automata-theoretic foundation.

## C. Example JSON Schema

We use the following JSON schema to measure the effect of canonicality and skipping decoding during generation for Table 2.

```
{
    "alias": "[a-z]{1,5}",
    "description": "[A-Z][a-z][0-9][\w\s,.!?]{1,3}",
    "type": "(Bug|Dark|Dragon|Electric|Fairy|Fighting|Fire|Flying|Ghost|Grass|Ground|Ice|Normal|Poison|Psychic|Rock|
        Steel|Stellar|Water)",
    "height_m": [0-9]{1,2}\.[0-9]{1},
    "weight_kg": [0-9]{1,3}\.[0-9]{1},
    "evolution_stage": "(Basic|Stage 1|Stage 2)",
    "legendary": "(true|false)",
    "abilities": [
        "(Overgrow|Blaze|Torrent|Shield Dust|Intimidate|Levitate|Pressure|Static)"(, "(Overgrow|Blaze|Torrent|Shield
            Dust|Intimidate|Levitate|Pressure|Static)"){0,6}
    ]
}
```

