# OpenReview forum: "Pitfalls, Subtleties, and Techniques in Automata-Based Subword-Level Constrained Generation"
_ICML.cc/2025/Workshop/TokShop — TokShop_

### Official Review · Reviewer_NgXm · 2025-06-07
**The paper discusses issues related to tokenization for constrained generation and proposes an implementation based on an automata framework.**

**Rating:** 7
**Confidence:** 2

**Review:**

This paper describes the implementation of a tool for constrained generation, with a particular focus on issues related to tokenization.
The presented implementation is based on an automata framework, by compiling automata from tokenizer configurations.

The paper discusses the following issues:

- preprocessing; where text preprocessing and normalization steps need to be considered, such as the the adding of leading spaces in SentencePiece.

- subword promotion; which is handled by transducing character sequences to subword tokens

- canonical filtering; which restricts the subword automaton to the canonical tokenization, and thus reduces the number of acceptable sequences.

I found the paper generally well written and very interesting; it discusses very thoroughly potential issues, clearly describing the background and motivation for the choices for the  implementation, while also providing a comparison to existing tools for constrained generation.

Comment:
Typo: Example ?? in line 391

---

### Official Review · Reviewer_A5xg · 2025-06-08
**Good attempt on constrained generation and illustrating the pitfalls**

**Rating:** 7
**Confidence:** 3

**Review:**

This work is an invaluable contribution to constrained generation in LLMs, meticulously detailing and addressing a collection of sensitive yet significant pitfalls in existing implementations. Its greatest strength is the automata-theoretic solution, which provides a sound, verifiable, and elegant treatment to difficult-to-combine sets of user constraints with tokenizer specificities.

Quality and Clarity: The paper excels in clarity of presentation and organization. Preprocessing and encouraging subwords, to cite an example, is well explained and renders difficult concepts easy to understand. The description of canonical filtering, its benefit to inference efficiency (coalescence), and inductive model bias conservation is particularly enlightening. The overview table of methods (Table 1) as well as sample code (Figure 5) greatly enhance readability as well as understanding.

Originality and Relevance: This paper is not the first to create automata for constrained generation, yet the specific emphasis in this paper on practical implementational pitfalls and optimizations is extremely innovative. The introduction and integration of canonicalization as an explicitly precomputable step into automata is a strong differentiator from the literature, with optimizations guaranteed at no heuristic cost. Problem instances considered, reflecting many found in production-quality libraries, highlight the work's practical significance, with stronger, yet more efficient, constrained decoding systems on offer. Its lightweight research-oriented implementation also provides encouragement for further investigation.

Pros

In-depth identification regarding the common pitfalls in today's libraries.

Solid automata-theoretic foundation with guaranteed correctness and efficiency.

Modular design allowing children to independently adjust the constraints.

Performance improvement as evidenced through canonical filtering, particularly the generation of JSON.

Easy-to-use interface which conceals the tokenizer complexities.

Cons
Empirical assessment is limited to generation in JSON form and acknowledges its dependency on specialized models as well as tasks, where generalizability of performance claims might be limited.

Canonical filtering, although beneficial, can lead to very large automata in some applications, impacting efficiency.


It is primarily tuned to conventional constraints and the LLAMA2 tokenizer but generalizability is claimed.


Overall, it is an excellent paper that well integrates theoretical foundations and practical considerations in LLM constrained generation. It provides excellent insights with a robust framework that will definitely influence research and development in the area in the future.

---

### Decision · Program_Chairs · 2025-06-10

Accept